# Physics-informed deep learning approach for modeling crustal deformation

Tomohisa Okazaki [1] ✉, Takeo Ito [2], Kazuro Hirahara[1] & Naonori Ueda[1]

The movement and deformation of the Earth's crust and upper mantle provide critical insights into the evolution of earthquake processes and future earthquake potentials. Crustal deformation can be modeled by dislocation models that represent earthquake faults in the crust as defects in a continuum medium. In this study, we propose a physics-informed deep learning approach to model crustal deformation due to earthquakes. Neural networks can represent continuous displacement fields in arbitrary geometrical structures and mechanical properties of rocks by incorporating governing equations and boundary conditions into a loss function. The polar coordinate system is introduced to accurately model the displacement discontinuity on a fault as a boundary condition. We illustrate the validity and usefulness of this approach through example problems with strike-slip faults. This approach has a potential advantage over conventional approaches in that it could be straightforwardly extended to high dimensional, anelastic, nonlinear, and inverse problems.

Geodetic observations made using different sensors and instruments, including global navigation satellite systems, have produced a significant amount of data on crustal deformation. Modeling such observed deformation is fundamental to understanding the mechanics of earthquake processes[1,2]. A dislocation model of a slip on earthquake faults is commonly used for the forward and inverse modeling of coseismic and postseismic deformation and earthquake cycles[3–5]. Over the past decades, analytical and semianalytical approaches have been developed for linear rheologies[6–8], and fully numerical methods have been constructed for complex structures, including nonlinear mechanical properties[9–11].

Recent advances in machine learning, especially deep learning techniques, have occurred due to the large amount of available data[12–14]. Applications in geophysics have been implemented by utilizing accumulating seismic records[15–17]. In contrast, deep learning has also promoted the application of machine-learning approaches to physical systems, specifically the solution of partial differential equations (PDEs). Automatic differentiation[18] developed for the optimization of neural networks (NNs) plays a central role in the efficient computation of derivatives[19,20]. Among them, physics-informed neural networks (PINNs) have been proposed for solving both the forward and inverse problems of PDEs in a unified way[21]. PINNs represent continuous solutions without discretization and can be trained to conform to a physical law by incorporating the target PDEs and boundary/initial conditions into loss functions. Because of their simple implementation and applicability to different problem types, PINNs have received significant attention in physics and engineering[22]. In geophysics, a seismic inversion method was developed based on the similarity between automatic differentiation and the adjoint-state method[23]. PINNs have been applied to synthetic models of seismic tomography[24] and full waveform inversions[25].

This study applies PINNs to dislocation models of crustal deformation. An essential characteristic of these models is that the displacement field is discontinuous across the fault surface and cannot be directly approximated by NNs that represent continuous functions. To resolve this difficulty, we set an appropriate coordinate system to separate the values on the two sides of the displacement discontinuity. This formulation enables precise modeling of crustal deformation, including near-fault locations. PINNs can be applied to complex structures and easily extended to high-dimensional, anelastic, and nonlinear problems, which serves as a potential advantage over conventional approaches as follows.

[1]RIKEN Center for Advanced Intelligence Project, Seika, Japan. [2]Graduate School of Environmental Studies, Nagoya University, Nagoya, Japan.
✉e-mail: tomohisa.okazaki@riken.jp

Analytical approaches use Green's functions (GFs) to study crustal deformation from arbitrary slip distributions. They provide continuous solutions and explicit dependencies on model parameters. Exact expressions of elastostatic GFs have been obtained for several crustal structures, such as homogeneous half-space[6], layered half-space[26], and layered spherical Earth[27]. Viscoelastic rheology is addressed using the correspondence principle. Deformation due to a finite fault is expressed as a convolution of GFs and slip distribution on the fault. The surface topography is addressed by a series expansion assuming a small slope[28–30]. However, the known analytical solutions have been limited to simple structures.

Semianalytical approaches have been developed to model composite rheologies and quasi-static earthquake cycles[7,8,31]. A curved fault is typically expressed as a sum of planar sub-faults because its GFs require a deliberate derivation[32]. The general topography is addressed by sophisticated formulations with full-space GFs[33]. However, the assumption of linear responses is a major limitation of these methods. Although a general representation of anelastic deformation using GFs was formulated for nonlinear quasi-static problems[34], GFs have only been derived in a homogeneous half-space[35].

Realistic problems are solved through fully numerical methods. In particular, the finite element method (FEM) is suitable for modeling complex tectonic settings such as subduction zones[2,10,36], mountain regions[37], and nonlinear rheologies[11]. PINNs share common advantages with FEM: topography and heterogeneity can be modeled, and nonlinear PDEs are implemented in a similar manner to linear PDEs[21]. Therefore, PINNs are frequently compared with FEM[38]. The FEM generates a discretized mesh, which restricts the model resolution. A large amount of memory required to store information at every grid point can make modeling complex structures difficult. In contrast, PINNs directly model a continuous field, which potentially achieves high accuracy by trained on arbitrarily large number of points. NNs can be trained with minibatch iteratively without storing all training points, which could be a computational advantage in modeling realistic large-scale problems. PINNs have another advantage in directly solving infinite domain problems without imposing boundary conditions on model boundaries; however, FEM can solve only finite domain problems and often requires modeling domains that are far larger than the target domains with careful treatment of boundary conditions to reduce boundary effects.

In a pioneering application of NNs to the forward modeling of crustal deformation[39], NNs were trained on simulation results of semianalytical methods[6,26] to interpolate solutions at arbitrary locations and model parameters such as source depth and viscosity. This approach considerably accelerates the estimation of deformation in simple problems to which semianalytical approaches can be applied, whereas PINNs can address complex problems without any existing solver despite a longer computational time.

## Results

### Physics-informed neural network modeling

We consider slips on strike-slip faults, a model that could be used to describe repeated destructive earthquakes. For simplicity, we assume infinitely long strike-slip faults in linear elastic media. By taking the $x$- and $z$-axes in the horizontal direction and the $y$-axis in the vertical direction, we suppose a displacement $u(x, y)$ to be parallel to and invariant in the $z$-direction. We denote a medium as $V$, a fault (dislocation surface, DS) as $\Sigma$, and the Earth's surface (free surface, FS) as $S$ (Fig. 1a). Because $u(x, y)$ is discontinuous across $\Sigma$, we use the polar coordinates $(r, \theta)$ whose branch cut is defined along $\Sigma$. In this way, $u(r, \theta)$ is continuous in the entire domain and the slip on $\Sigma$ can be represented as a constraint between $\Sigma_+$ and $\Sigma_-$ (Fig. 1b).

PINN modeling consists of three building blocks (Fig. 1c). First, an NN surrogates a continuous displacement field $u(r, \theta)$. Second, derivatives of the output $u$ with respect to the input variables $(r, \theta)$ are

evaluated using automatic differentiation[18]. Finally, a loss function $L$ is defined by the sum of the squared residual of the governing equation and boundary conditions: the equilibrium of linear elasticity $L_{PDE}$, the displacement discontinuity on the fault $L_{DS-u}$, the traction continuity on the fault $L_{DS-T}$, and the free-surface condition on the Earth's surface $L_{FS}$. The NN parameters are updated to decrease the loss function $L$ using a stochastic gradient method. In this study, training is iterated until $L < 10^{-6}$ is satisfied for fixed grid points. See 'Methods' for the details of the NN architecture, mathematical expressions of $L$, and optimization procedure.

### Modeling applications

We first consider vertical faults in a homogeneous half-space, for which analytical solutions are known. In homogeneous media, deformation is independent of a shear modulus $\mu$. The following three models are considered. Models 1A and 1C represent a surface fault (Fig. 2a, left). Model 1B represents a buried fault extending from the locking to an infinite depth (Fig. 2a, right), which has been used for strain accumulation along a plate boundary with a relative motion during interseismic periods[40]. By taking $\theta = 0$ upward, the surface and buried faults are expressed by branch cuts at $\theta = 0$ and $\theta = \pi$, respectively. Fault slips are uniform in Models 1A and 1B, whereas a distributed slip of a tapered stress is assumed in Model 1C (Fig. 2b). In the following modeling, all quantities are normalized by characteristic scales: spatial coordinates $x$ and $y$ by a fault depth $d$, displacement $u$ by a maximum slip amount $s_0$, and shear modulus $\mu$ by a reference value $\mu_0$. For example, strain is expressed in the unit of $s_0 / d$. The surface displacement and strain $\varepsilon_{xz}$ are shown in Fig. 2c, d with analytical solutions[3] (2-D displacement and strain are shown in Supplementary Fig. 1). The root-mean-square errors (RMSEs) are shown in Table 1. PINN solutions are generally accurate and the RMSE is sufficiently smaller than typical $u$ and $\varepsilon$ values. The RMSE is larger in $u$ than in $\varepsilon$ by approximately five times. This is because the governing PDE constrains not $u$ itself but its spatial derivatives. Deformation is localized in the distributed slip (Model 1C) compared to that in the uniform slip (Model 1A), which results in a higher strain near the fault.

PINNs have advantages in the modeling of complex crustal structures. In particular, continuous geometry and changes in mechanical properties can be expressed without any approximation such as series expansion and discretization. Figure 3a shows the vertical section of an example dislocation model; a fault and the Earth's surface are curved, and the mechanical property varies continuously in space. Two slip distributions, a uniform slip in Model 2A and a distributed slip identical to Model 1C (Fig. 2b) in Model 2B, are considered on the fault.

The obtained strain fields $\varepsilon_{yz}$ are shown in Fig. 3b (2-D displacement and strain are shown in Supplementary Fig. 2). The strain field in Model 2A is continuous and independent of the fault surface (dislocation surface) and was concentrated at the lower tip of the fault (dislocation line). Thus, crustal deformation for a uniform slip is completely determined by a dislocation line, which can be observed for different fault geometries (Supplementary Fig. 3). This property is well known for a plane fault in a homogeneous half-space[3], whereas the present result applies to a curved fault in a heterogeneous medium. In contrast, the strain field for Model 2B is discontinuous on the fault surface. The surface displacement and strain are shown in Fig. 3c, d with FEM solutions (see Supplementary Text 1 for the FEM modeling). The discrepancy in these models is larger than that in Models 1A–1C but not more than twice that in Model 1C (Table 1). In Model 2B, a high strain is distributed on the fault surface (Fig. 3b), which results in localized displacement $u$ (Fig. 3c) and high strain near the fault on the Earth's surface (Fig. 3d).

The Earth's crust is composed of various rock types with different mechanical properties. Strain accumulates at material boundaries where earthquakes tend to occur. Some earthquake faults are located

within damage zones extending to a considerable depth. Therefore, the modeling of discontinuous media is of practical importance. In this study, a displacement field is modeled with two NNs in the individual material regions ($V_1$ and $V_2$). Two terms $L_{MB\text{-}u}$ and $L_{MB\text{-}T}$ are added to the loss function $L$ to impose appropriate conditions on the material boundary $B$. See 'Methods' for more details.

Here, we consider that the mechanical property changes discontinuously on either side of a buried vertical fault. Model 3A has a discontinuity across the fault and Model 3B has a compliant fault zone (Fig. 4a). The contrast in shear modulus is two in both models. The obtained strain fields $\varepsilon_{xz}$ are shown in Fig. 4b (2-D displacement and strain are shown in Supplementary Fig. 4). Strain is discontinuous across the material boundary $B$. The surface displacement and strain are shown in Fig. 4c, d with analytical solutions[3], and RMSEs are shown in Table 1. PINN solutions are accurate in Model 3A but exhibit a systematic overestimation at a long distance from the fault in Model 3B. This would be because the material boundary $B$ isolates $V_2$ from the fault $\Sigma$ on which a displacement discontinuity (i.e. Dirichlet boundary condition) is imposed, which leads to error accumulation at a long distance. This suggests that multiple material discontinuities can complicate PINN's convergence property.

## Computational costs

Table 1 summarizes model structures and computational costs on a single CPU (Intel Core i7, 3.60 GHz, 4 cores, 8 processors, and 16 GB memory) of example problems in this study. We note that NNs consist of 8 hidden layers with 40 nodes and the batch size of training is 256 in

$V$ and 64 on $\Sigma$, $S$, and $B$ in all problems (see 'Methods' for details). The number of iterations required to achieve the desired precision ($L < 10^{-6}$) has significant dependence on model structures. The geometry and mechanical properties had minor effects (Model 2A), whereas the distributed fault slip led to many iterations (Models 1C and 2B). This might be related to the property that strain only depends on the location of dislocation lines for uniform slips, as discussed previously. The material discontinuities (Models 3A and 3B) did not significantly increase the number of iterations but increased computational time per iteration by approximately 1.6 folds because the use of two NNs doubles the number of NN parameters. The column 'Transfer' indicates that the trained NN parameters on similar but simple problems are used as initial weights of the target problems (see 'Methods' for details). In experiments, training without transfer increased computational time by 1.15, 1.38, 4.68, and 2.13 folds in Models 1C, 2B, 3A, and 3B, respectively. This indicates that the transfer of NN parameters is particularly effective for discontinuous material problems.

The computational time in the FEM modeling is 256 and 715 s for Models 2A and 2B, respectively, which is significantly shorter than that in the PINN modeling (Table 1). Here, the number of cells and nodes in the FEM models is 790,392 and 138,110, respectively (Supplementary Text 1). Computational cost is currently a common challenge of PINN forward modeling[41,42]. Fast and stable algorithms for PINN optimization should be investigated. In FEM, sophisticated mesh generation schemes including fault interfaces have been developed[43]. Knowledge and experience on conventional solvers would play an essential role in the progress.

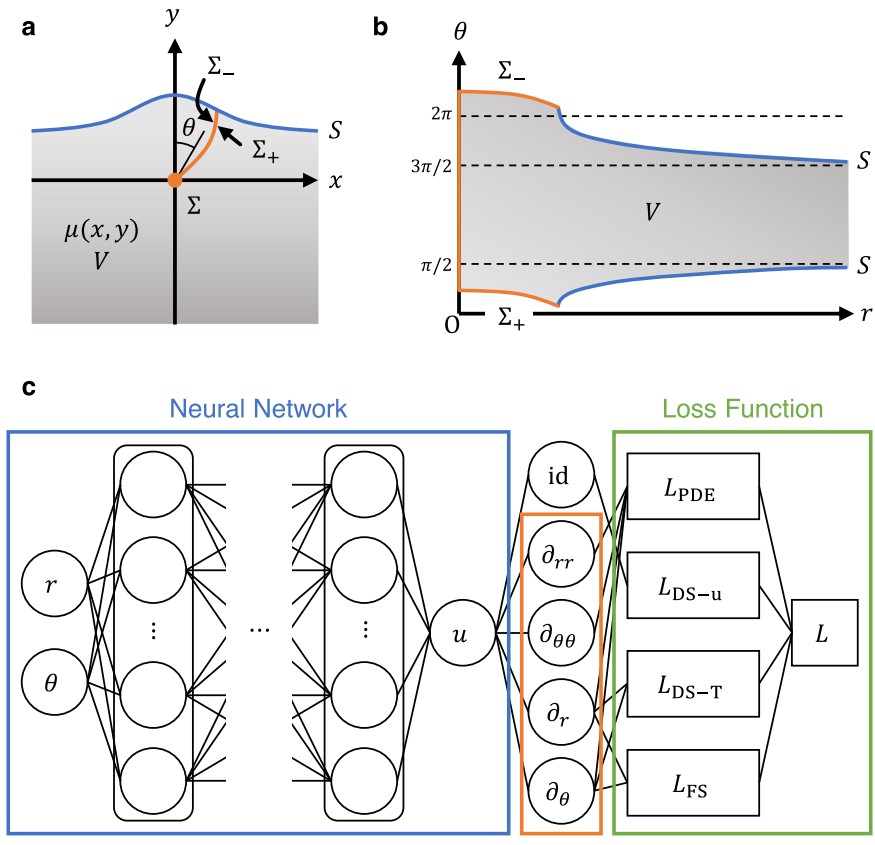

**Fig. 1 | Physics-informed neural network (PINN) modeling of antiplane dislocations. a** Cross-sectional view of an infinitely long strike-slip fault model. Shaded region, orange line, and blue line represent elastic medium $V$ of shear modulus $\mu(x,y)$, fault $\Sigma$, and Earth's surface $S$, respectively. $\Sigma_+$ and $\Sigma_-$ represent the two sides of $\Sigma$. The fault moves perpendicular to the page. **b** Polar coordinate representation of the model region. **c** PINN structure. A neural network surrogates a displacement $u(r, \theta)$ in the polar coordinates. Derivatives of $u$ are calculated using automatic differentiation and constitute a loss function $L$.

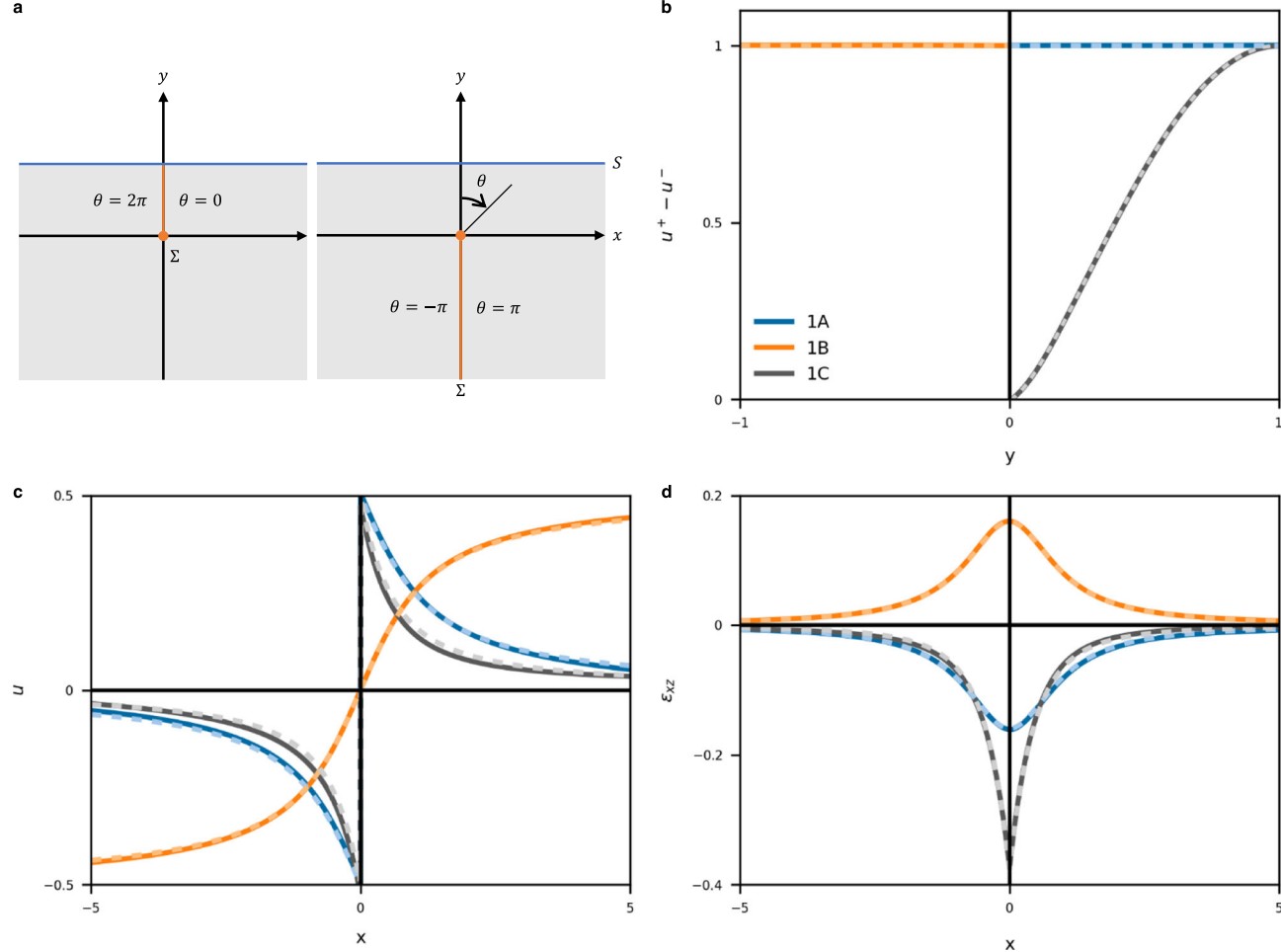

**Fig. 2 | Model and estimated results in a homogeneous half-space. a** Model structures. The orange and blue lines represent fault $\Sigma$ and Earth's surface $S$, respectively. Surface (left) and buried (right) faults are distinguished by the branch cut. **b** Assumed displacement discontinuities on the fault. **c** Estimated surface displacements. **d** Estimated surface strains. In (**b**)–(**d**), blue, orange and gray lines indicate Models 1A, 1B, and 1C, respectively. Solid and dashed lines indicate PINN and analytical solutions, respectively.

**Table 1 | Model structures, root-mean-square errors (RMSEs), and computational costs**

| Model | Transfer | Fault | RMSE ($u$) | RMSE ($\varepsilon$) | Iterations | Time (s) | $T/I$ (ms) |
|---|---|---|---|---|---|---|---|
| 1A | – | Surface | 6.733e−3 | 1.213e−3 | 22,875 | 825 | 36.07 |
| 1B | – | Buried | 3.372e−3 | 6.257e−4 | 19,487 | 705 | 36.18 |
| 1C | 1A | Surface | 1.409e−2 | 2.732e−3 | 140,229 | 4892 | 34.89 |
| 2A | – | Surface | 2.557e−2 | 3.727e−3 | 24,612 | 894 | 36.32 |
| 2B | 2A | Surface | 1.947e−2 | 3.957e−3 | 436,583 | 15,899 | 36.42 |
| 3A | 1B | Buried | 5.706e−3 | 1.013e−3 | 7803 | 466 | 59.72 |
| 3B | 1B | Buried | 1.523e−2 | 2.636e−3 | 37,583 | 2185 | 58.14 |

$u$ displacement, $\varepsilon$ strain, $T/I$ time per iteration.

## Discussion

This study focused on a dislocation model of strike-slip faults; however, PINNs can model dip–slip faults and quasi-static processes with a slight increase in the input and output variables of NNs. Inverse problems can be formulated by adding a data misfit term to the loss function; the simple implementation of inversion analyses is a major advantage of PINNs over conventional linear solvers. In fact, PINNs have currently succeeded more in inverse modeling than in forward modeling[22]. Because geophysical data are typically noisy, a Bayesian approach[44] may be required for stable inversion. PINNs can also be applied to general rheologies, such as viscoelasticity, poroelasticity, and power law creep, by changing the loss term based on the governing equation. The difference in regional rheologies (e.g., elasticity in the crust and viscoelasticity in the upper mantle) can be treated by defining loss functions in each subregion, which is similar to discontinuous media.

This study presented successful applications of PINNs for modeling crustal deformation in antiplane problems with simple NN

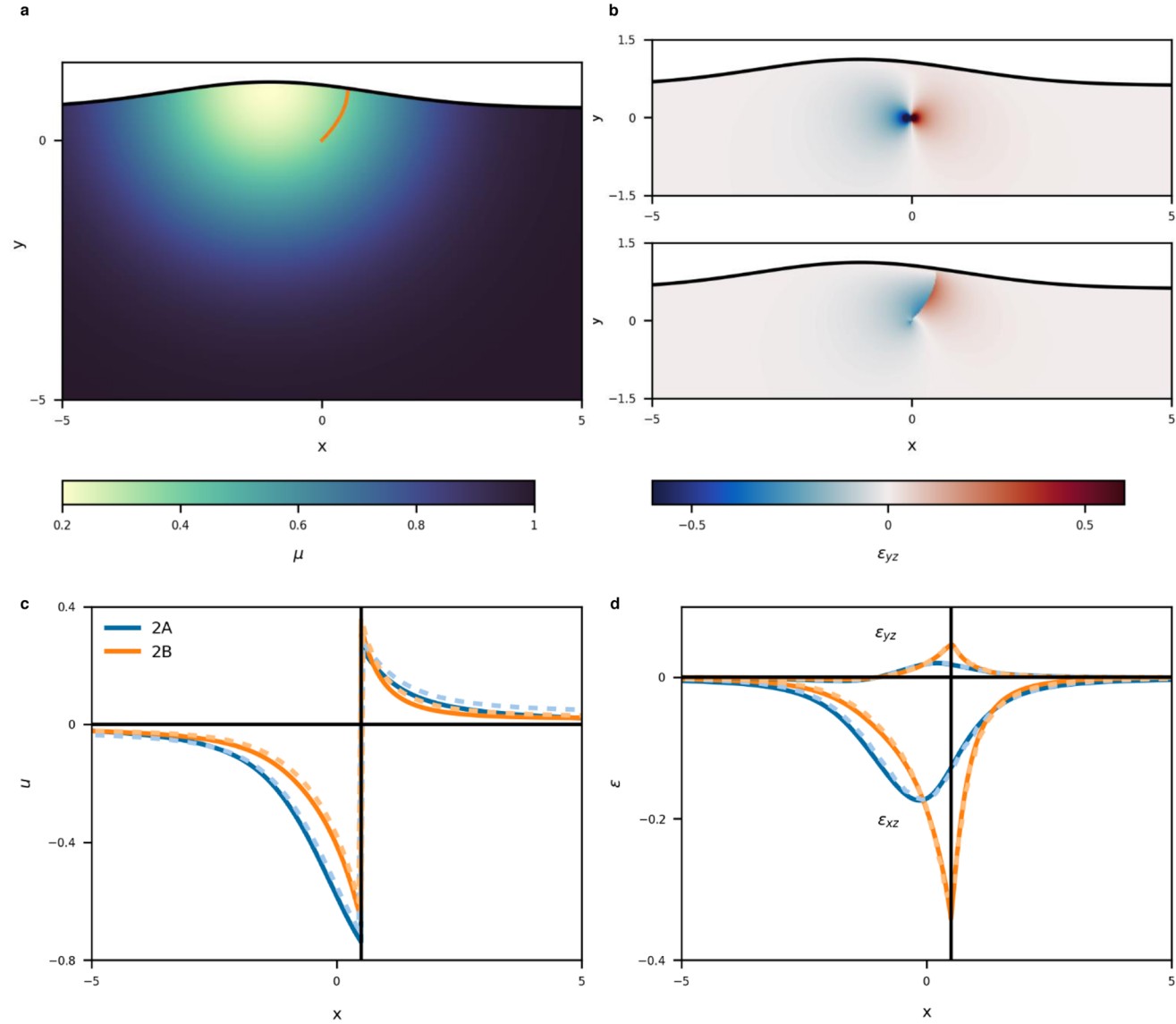

**Fig. 3 | Model and estimated results of a curved, heterogeneous structure.**
**a** Model structure. Colors represent the shear modulus. The orange curve represents the fault surface. **b** Estimated strains for Models 2A (top) and 2B (bottom).

**c** Estimated surface displacements. **d** Estimated surface strains. In (**c**) and (**d**), blue and orange lines indicate Models 2A and 2B, respectively. Solid and dashed lines indicate PINN and FEM solutions, respectively.

architecture and optimization procedure. However, realistic problems require large and higher dimensional model space, and various material properties and rheologies require an increasing number of corresponding NNs. It has been recognized that PINNs can sometimes converge to erroneous solutions in time-dependent modeling[45,46]. These factors would incur more computational costs for optimization of the loss function to achieve sufficient accuracy. Understanding the method for stable and fast optimization is key for the application of PINNs to large-scale geophysical problems. PINNs[21] are newcomers to machine learning, and studies aimed at realizing faster and more efficient optimization have been accelerating[45–48]. Therefore, our proposed approach based on PINNs may be a powerful tool for realizing a wide variety of modeling applications in crustal deformation.

## Methods
### Dislocation model
In this study, we consider antiplane dislocations, which model infinitely long strike-slip faults. By taking the $x$- and $z$-axes in the horizontal direction and the $y$-axis in the vertical direction, we suppose that a displacement $u(x, y)$ is parallel to and invariant in the $z$-direction. We

denote a medium as $V$, a fault (dislocation surface, DS) as $\Sigma$, and the Earth's surface (free surface, FS) as $S$ (Fig. 1a). The normal vectors to $\Sigma$ and $S$ are denoted by $\mathbf{n}^{DS} = (n_x^{DS}, n_y^{DS}, 0)$ and $\mathbf{n}^{FS} = (n_x^{FS}, n_y^{FS}, 0)$, respectively. In an isotropic linear elastic medium, the system of governing equations is given by[3]

$$\mu \nabla^2 u + \nabla \mu \cdot \nabla u = 0 \text{ in } V, \tag{1}$$

$$u^+ - u^- = s \text{ on } \Sigma, \tag{2}$$

$$\boldsymbol{\sigma}^+ \cdot \mathbf{n}^{DS} = \boldsymbol{\sigma}^- \cdot \mathbf{n}^{DS} \text{ on } \Sigma, \tag{3}$$

$$\boldsymbol{\sigma} \cdot \mathbf{n}^{FS} = 0 \text{ on } S, \tag{4}$$

where $\mu$ is the shear modulus, $s$ is the slip on $\Sigma$, $\boldsymbol{\sigma}$ is the stress tensor, and the superscripts $+$ and $-$ represent the opposite sides of $\Sigma$. The first equation represents the equilibrium equation of linear elasticity, the second represents the displacement discontinuity on the

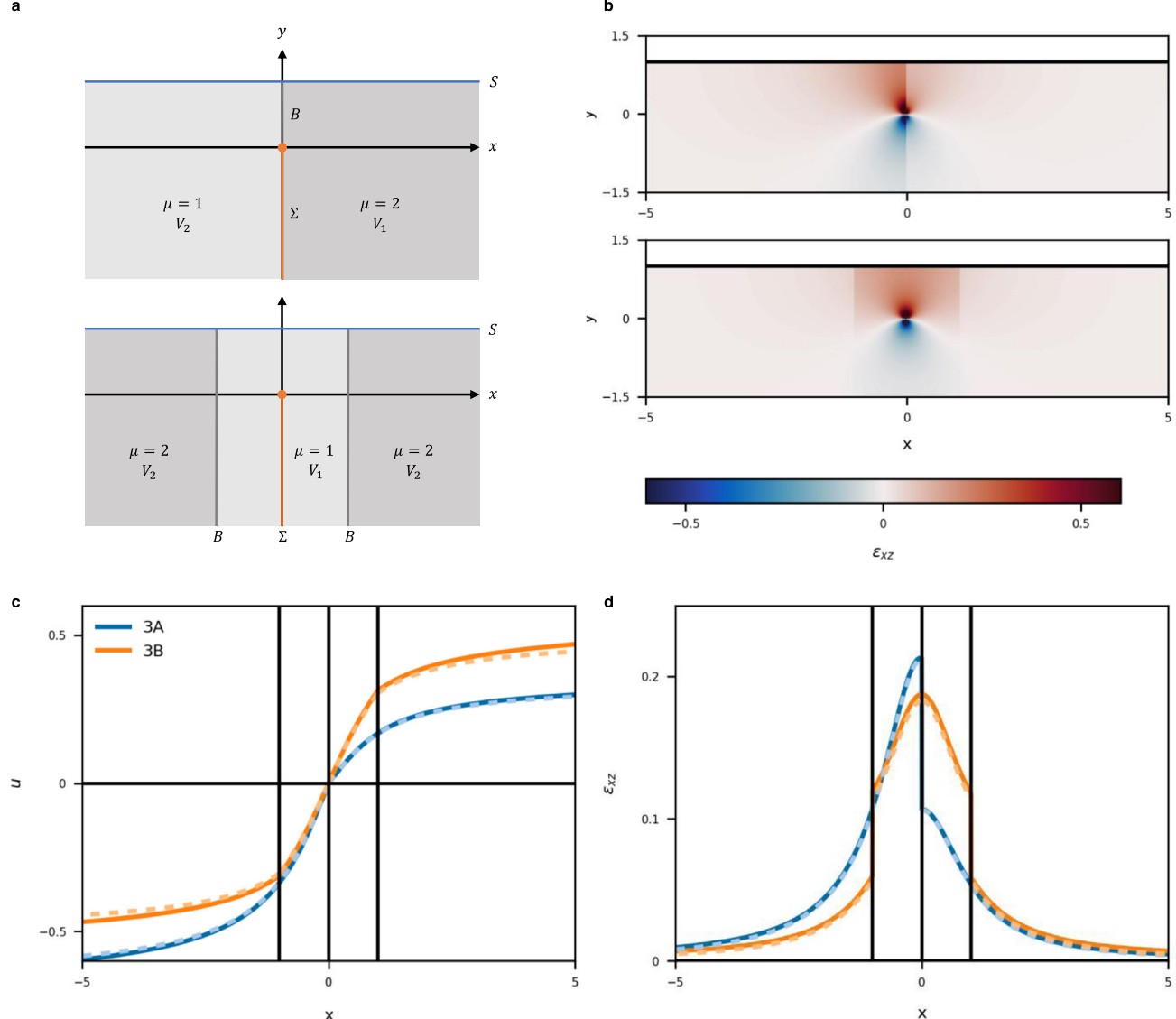

**Fig. 4 | Model and estimated results of discontinuous structures. a** Model structures of Models 3A (top) and 3B (bottom). Shaded regions represent elastic media $V_1$ and $V_2$ of shear modulus $\mu$. The orange, blue, and gray lines represent fault $\Sigma$, Earth's surface $S$, and material boundary $B$, respectively. **b** Estimated strains for Models 3A (top) and 3B (bottom). **c** Estimated surface displacements. **d** Estimated surface strains. In (**c**) and (**d**), blue and orange lines indicate Models 3A and 3B, respectively. Solid and dashed lines indicate PINN and analytical solutions, respectively.

fault, the third represents the traction continuity on the fault, and the fourth represents the free-surface condition on the Earth's surface.

Next, we consider that two regions $V_1$ and $V_2$ with shear moduli $\mu_1$ and $\mu_2$, respectively, are in contact with a material boundary (MB) $B$ (Fig. 4a). By denoting the displacements in $V_1$ and $V_2$ as $u_1$ and $u_2$, respectively, the boundary conditions are expressed as

$$u_1 = u_2 \, \text{on} \, B, \tag{5}$$

$$\boldsymbol{\sigma}_1 \cdot \mathbf{n}^{\text{MB}} = \boldsymbol{\sigma}_2 \cdot \mathbf{n}^{\text{MB}} \, \text{on} \, B, \tag{6}$$

where $\mathbf{n}^{\text{MB}} = (n_x^{\text{MB}}, n_y^{\text{MB}}, 0)$ is the normal vector to $B$. They represent the displacement and traction continuity at the material boundary, respectively.

## Neural network modeling

The displacement field $u$ is modeled by NNs. $u(x, y)$ is discontinuous across a fault surface $\Sigma$ (dislocation surface) and the stress can diverge at the fault tip (dislocation line), which prevents NNs from generating an accurate approximation. We therefore use the polar coordinate system $(r, \theta)$ whose pole is located at the dislocation line, and define a branch cut along the dislocation surface. A curved fault is expressed by a branch cut as a function of $r$. Modeling of $u(r, \theta)$ by NNs separates the coordinate values of the two sides of $\Sigma$ (Fig. 1b), which results in the accurate modeling of displacements near fault surfaces.

A material boundary induces discontinuity not in displacements but in strains (derivatives of $u$). Therefore, it is difficult to approximate displacement using a single NN. We train the NNs in individual material regions and impose boundary conditions to ensure consistency between them. This is similar to the domain decomposition introduced to accelerate the convergence[49].

## Loss function

A loss function is defined as the residuals of the governing equations and the boundary conditions. Using the stress–strain relation $\sigma_{xz} = \mu u_x / 2$ and $\sigma_{yz} = \mu u_y / 2$ for antiplane strains, the individual loss terms in the polar coordinates are written as

$$L_{\text{PDE}} = [r^2 u_{rr} + r u_r + u_{\theta\theta} + \mu^{-1}(r^2 \mu_r u_r + \mu_\theta u_\theta)]^2, \quad (7)$$

$$L_{\text{DS-u}} = (u^+ - u^- - s)^2, \quad (8)$$

$$L_{\text{DS-T}} = r^2 [\mu^+ (n_x^{\text{DS}} u_x^+ + n_y^{\text{DS}} u_y^+) - \mu^- (n_x^{\text{DS}} u_x^- + n_y^{\text{DS}} u_y^-)]^2, \quad (9)$$

$$L_{\text{FS}} = r^2 \left( n_x^{\text{FS}} u_x + n_y^{\text{FS}} u_y \right)^2, \quad (10)$$

$$L_{\text{MB-u}} = (u_1 - u_2)^2, \quad (11)$$

$$L_{\text{MB-T}} = r^2 \left[ \mu_1 \left( n_x^{\text{MB}} u_{1x} + n_y^{\text{MB}} u_{1y} \right) - \mu_2 \left( n_x^{\text{MB}} u_{2x} + n_y^{\text{MB}} u_{2y} \right) \right]^2, \quad (12)$$

where $u_x = \sin\theta u_r + r^{-1}\cos\theta u_\theta$ and $u_y = \cos\theta u_r - r^{-1}\sin\theta u_\theta$. The subscripts of $u$ represent partial derivatives (e.g. $u_x = \partial u / \partial x$ and $u_{rr} = \partial^2 u / \partial r^2$). Automatic differentiation[18] enables exact and efficient calculations of partial derivatives. The powers of $r$ are multiplied to remove singular values at the origin. The total loss function for continuous media is given by

$$L = L_{\text{PDE}} + L_{\text{DS-u}} + L_{\text{DS-T}} + L_{\text{FS}}, \quad (13)$$

and that for discontinuous media is given by

$$L = L_{\text{PDE}} + L_{\text{DS-u}} + L_{\text{DS-T}} + L_{\text{FS}} + L_{\text{MB-u}} + L_{\text{MB-T}}. \quad (14)$$

## Optimization

We use the same NN structure for all examples by mainly following the original work[21]. Fully connected feedforward NNs consisting of 8 hidden layers with 40 nodes are used. The activation functions are the hyperbolic tangent function in the hidden layers and the identity function in the output layer. Xavier's initial value[50] is used as initial NN parameters. Moreover, when two or more similar problems are considered, the trained NN parameters on a simple problem are transferred to the initial NN parameters of complex problems. This can be interpreted as a variant of the curriculum learning[45], which aims at accelerating and stabilizing PINN optimization. The correspondences are listed in Table 1. The NN parameters are updated using the gradient-based algorithm Adam[51] with standard learning rates ($\eta = 10^{-3}$, $\beta_1 = 0.9$, and $\beta_2 = 0.999$). PINNs do not require training data, and loss functions are calculated at arbitrary points in the model domain, which are called collocation points. The range of collocation points is set as $-5 \le x \le 5$, $-5 \le y$, and the upper bound defined by the Earth's surface. The batch size is set to 256 in $V$ and to 64 on $\Sigma$, $S$, and $B$. Collocation points are independently sampled in each training step.

The distribution of collocation points during training can have a significant influence on the model performance[38]. The influence is inspected by observing the spatial distribution of a residual $R = r^2 u_{rr} + r u_r + u_{\theta\theta} + \mu^{-1}(r^2 \mu_r u_r + \mu_\theta u_\theta)$, the mean square of which is a loss term, $L_{\text{PDE}}$ (Supplementary Fig. 5a–d). When an NN is trained on collocation points sampled from a uniform distribution, the residuals exhibit higher values near the fault. If collocation points are sampled from a probability distribution that concentrates on the fault, the residual is uniformly distributed. Therefore, concentrated sampling is

used in this study. Examples of collocation points in the analyzed models (Figs. 3 and 4) are shown in Supplementary Fig. 5e, f.

Because the values of a loss function can vary considerably with the sampled collocation points, fixed fine grids are prepared for model evaluation. The grid intervals are set to 0.1 in $V$ and to 0.01 on $\Sigma$, $S$, and $B$. Minibatch training is iterated until the following conditions are satisfied: if the training loss $L_{\text{tra}}$ on collocation points is less than $10^{-6}$, the validation loss on the fixed grids $L_{\text{val}}$ is calculated; if $L_{\text{val}}$ is also less than $10^{-6}$, training is finished (Supplementary Fig. 6).

## Data availability

No data were used in this study.

## Code availability

Source programs of PINN modeling are available in Supplementary Software 1. FEM solutions were generated using the open-source Python package, PyLith[43].

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

## Author contributions

T.O. designed the study, carried out the numerical modeling with PINN, and prepared the manuscript. T.I. carried out the numerical modeling with FEM. K.H. and N.U. advised the project. All authors discussed the results in the article.

## Competing interests

The authors declare no competing interests.
