## [Peer Review File · Nature Communications]

Physics-Informed Deep Learning Approach for Modeling Crustal DeformationREVIEWER COMMENTS

Reviewer #1 (Remarks to the Author):

This article provides a great application of PINNs in geophysics. The authors demonstrate how PINNs, a meshless solver can be used seamlessly to solve crustal deformation after an earthquake. This approach opens new doors and can potentially in the future replace complex finite element codes that are computationally intensive.

My major concern is the lack of a reference solution to test PINN's results for the heterogeneous fault models. The authors provide a reference exact solution for the homogeneous fault model to show the accuracy of the PINN solution, but this argument by no means guarantee that their method will still be accurate for the more complex heterogeneous cases. In fact, accuracy is one of the main issues with PINNs and without a reference model it is impossible to establish a strong and general argument in favor of PINN. Please pick a method (Finite difference or Finite element) and show how PINNs compare to mainstream numerical schemes.

I suggest that the authors discuss and compare the computational costs of PINN in their work with other common numerical schemes such as Finite Differences or Finite Elements. How long does it take to solve each of the problems in this paper with PINNs versus other methods assuming the same computational resources (CPUs, GPUs etc.).

Please elaborate on how you chose the ideal network architecture? How does the design of the NN (number of layers and number of neurons in each layer) affect accuracy? How does the NN's architecture change from homogenous fault model to the heterogeneous model?

I suggest authors to test a fault model with discontinuous material heterogeneities on each side of the fault. The models discussed are relatively smooth. Can PINN handle these sorts of problems with a desired accuracy?

Line 49 "PINNs represent continuous solutions without discretization and can be trained without observational data by incorporating the target PDEs and boundary/initial conditions into loss functions." This statement is not accurate. For inverse problems PINNs do need observational data, just like any other techniques.

Line 89 ". In contrast, PINNs directly model a continuous field, which potentially achieves high resolution." This statement is incorrect and needs a clear reference or discussion. Modeling via continuous functions does not automatically guarantee any improvement on the resolution, assuming a fix amount of given information.

Line 156 "Thus, PINNs have several advantages over conventional methods for modeling crustal deformation." The manuscript does not clearly show or support this claim.

Line 157 "However, larger-scale realistic problems with material discontinuities and rheologies require an increasing number of corresponding NNs, which incur more computational costs for..." Why more NNs? why not NNs with larger number of parameters? Please provide a more detailed discussion here.

Reviewer #2 (Remarks to the Author):

The paper "Physics-Informed Deep Learning Approach for Modeling Crustal Deformation" by Tomohisa Okazaki, Kazuro Hirahara, and Naonori Ueda describes a new method for calculating stresses due to slip on faults. They propose using physics-informed neural networks (PINNs) trained directly on the underlying physical equations. They present applications for slip on faults embedded in heterogeneous elastic bulk and discuss the advantages and potential applications of this technique.

Overall I find this an important and timely contribution. The advantages of PINNs discussed in the paper (its ability to approximate continuous functions in heterogeneous media, handle complex geometries, and include non-elastic behavior) are indeed remarkable, and the authors effectively describe these aspects. However, some aspects of the paper can be improved to make it more accessible and useful to the readers: a different structure, more detail on the implementation, and a mention of the computation costs. The latter in particular is an important aspect to evaluate whether PINNs are a strong alternative to FEM. With these changes, I anticipate that the article will be suitable for publication in Nature Communications.

General Comments:

1) Structure. Since this is a methods paper, I recommend placing the Methods section in the main text and not in the extended information section. While reading the main text, I thought it was missing key information, such as the underlying equations and comparison to analytical solutions. This material is in the extended information, but it is central to the paper and I imagine that anyone who reads the article will want to read it as well. In terms of figure, I suggest placing Fig. 5 and perhaps Fig. 3 also in the main text.

2) Information on computational costs. To understand if PINNs are competitive compared to FEMs, an idea of the computational costs would be very useful: how long does training take, and how does it compare to using FEM directly? An estimate of the run times, with details of the hardware (e.g. CPUs or GPUs?) would be very valuable.

A related question is the cost of training several NNs for separate spatial domains (e.g. Figure 2). Does it simply scale linearly with the number of domains? This is important to understand if one wants to model complex cases with a large number of discontinuities.

3) More detail on how the model is constructed would help. For example:

* Ln. 344-351: please explain the choice of network architecture, activation functions, and batch sizes. Have you experimented with other options? Are these values standard for PINNs developed for other applications (if so, add references)? Is there any theoretical reason or a rule of thumb to choose these values?

* How do initial weights affect the time needed to train the network, and how did you choose them?

* What learning rate did you use for optimization and how did you choose it?

Specific Comments:

Ln. 101-102, and figures: Please describe the length scales and other normalizations, both in the text and in figures. Values for shear modulus and strain are unphysical. Since this is linear elasticity it all works out, but it can be confusing - a brief explanation would help.

Ln. 106: please describe the slip distribution for the “distributed slip” model.

Ln. 96. “We consider a model for strike-slip faults that repeatedly cause destructive earthquakes in inland areas.”: this sounds too specific a description for your problem setup; and for a moment I thought you were modeling seismic cycles. The fault could be creeping, or produce earthquakes that are not repeated, or not be inland. I would suggest phrasing as follows: “We consider slip on a strike slip fault, a model that could be used to describe repeated destructive earthquakes...”

Reviewer #3 (Remarks to the Author):

The authors propose using a powerful new tool of machine learning — physics-informed neural networks (PINNs) — to solve basic problems in crustal deformation. Attractive features are that analytical or numerical (e.g., finite element) solutions need not be constructed, only rudimentary information on physical properties and source parameters. They present two simple examples, one with a dislocation on a curved fault within a continuously varying heterogeneous medium, the other with a planar fault within a discontinuously varying medium. Both examples are relevant, and the authors demonstrate many desirable properties of the PINN solution.

Although I’m not an expert on machine learning, I find the present approach to be exciting and agree that it would open the door to new applications and change the way that we approach existing problems. This is especially true for the extension to quasi-static problems or any problem with time dependence (e.g., poroelastic problems) in addition to static problems. However, I feel that one issue must be addressed for this study to be publishable. That is to consider adding an example with a quasi-static deformation problem. This is because a previous contribution (DeVries et al., 2017) used a neural network approach for this more sophisticated problem, and you could take the opportunity to express how PINNs could do better or are at least a strong alternative. Once this issue is addressed I feel the study would be publishable in Nature.

Minor comments

Line 35 and lines 83-85. For fully numerical methods you could also refer to Langer et al. (2019).

Lines 41-43. “These techniques extract latent relations from observational data without knowing the physical laws, and their applications to problems with limited data have not been successful. “ I don’t quite know what you mean here, as the machine learning applications to seismic event detection have been very successful. Maybe just say, alternate applications to solving physics-based problems are not as amenable to these techniques.

All applications appear to be 2D (infinitely long faults). How would the computational cost increase if you did 3D problems?

References

DeVries, P. M. R., T. B. Thompson, and B. J. Meade (2017), Enabling large-scale

viscoelastic calculations via neural network acceleration, *Geophys. Res. Lett.*, 44, 2662–2669, doi:10.1002/2017GL072716.

Langer, L., Gharti, H. N., and Tromp, J., Impact of topography and three-dimensional heterogeneity on coseismic deformation, *Geophys. J. Int.*, Volume 217, Issue 2, May 2019, Pages 866–878, <https://doi.org/10.1093/gji/ggz060>

Response to reviewers' comments on

“Physics-Informed Deep Learning Approach for Modeling Crustal Deformation”

Reviewer #1

My major concern is the lack of a reference solution to test PINN's results for the heterogeneous fault models. The authors provide a reference exact solution for the homogeneous fault model to show the accuracy of the PINN solution, but this argument by no means guarantee that their method will still be accurate for the more complex heterogeneous cases. In fact, accuracy is one of the main issues with PINNs and without a reference model it is impossible to establish a strong and general argument in favor of PINN. Please pick a method (Finite difference or Finite element) and show how PINNs compare to mainstream numerical schemes.

Thank you for a critical comment on the validation of PINNs. We calculated FEM solutions for the heterogeneous fault models in Figure 3. We compared PINN solutions with analytical solutions in Figures 2 and 4.

I suggest that the authors discuss and compare the computational costs of PINN in their work with other common numerical schemes such as Finite Differences or Finite Elements. How long does it take to solve each of the problems in this paper with PINNs versus other methods assuming the same computational resources (CPUs, GPUs etc.).

PINN modeling is generally known to take more time than common solvers for now (references 40 and 41) and we consider that development of efficient algorithms is necessary to be comparable. Instead, we compared computational time between various target problems (Table 1), which clarified which factors increase computational time significantly. We mentioned computational time in the last paragraph of the original manuscript, but discussed it in more detail in the ‘Computational costs’ section (line 234–239).

In contrast, PINN has a potential advantage in memory consumption on large-scale problems. In FEM, the number of mesh grids increases considerably with model size and dimension. In PINN, although the number of iterations should be increase considerably, NNs are trained with minibatch and training points need not be stored. Therefore, memory consumption would not increase significantly. We discussed it in line 84–90.

Please elaborate on how you chose the ideal network architecture? How does the design of the NN (number of layers and number of neurons in each layer) affect accuracy? How does the NN's architecture change from homogenous fault model to the heterogeneous model?

In this study, we fixed a basic NN design on the basis of the original work (Raissi et al., 2019). In

particular, the NN architecture is the same in all examples. This implies stability of the performance against target problems. We consider that the search for an optimal design is important for technological progress in PINNs, but in this study, the demonstration of applicability without fine-tuning would be beneficial to state that PINNs are accessible to researchers who have a little experience on deep learning. We added explanations of the NN architecture in Figure 1 and the ‘Methods’ section (line 331–340).

I suggest authors to test a fault model with discontinuous material heterogeneities on each side of the fault. The models discussed are relatively smooth. Can PINN handle these sorts of problems with a desired accuracy?

We tested this model and found that it was indeed harder than the previous model to achieve a desired accuracy. We presented the results and discussed implications in Figure 4.

Line 49 “PINNs represent continuous solutions without discretization and can be trained without observational data by incorporating the target PDEs and boundary/initial conditions into loss functions.” This statement is not accurate. For inverse problems PINNs do need observational data, just like any other techniques.

We considered forward modeling in this sentence, but inverse modeling was surely described in the previous sentence. We rewrote the sentence to cover both cases (line 44–46).

Line 89 “. In contrast, PINNs directly model a continuous field, which potentially achieves high resolution.” This statement is incorrect and needs a clear reference or discussion. Modeling via continuous functions does not automatically guarantee any improvement on the resolution, assuming a fix amount of given information.

Our consideration is as follows. In FEM, once meshes are generated, the resolution is determined by mesh sizes. In PINN, a neural network can directly express continuous functions with many model parameters. In typical supervised learning, we have finite training data, whereas collocation points of PINN can be arbitrarily large (we can choose any point in the model domain). Therefore, information of PDE can be arbitrarily increased by training on more and more collocation points. We modified the sentence to clarify this point (line 84–88).

Line 156 “Thus, PINNs have several advantages over conventional methods for modeling crustal deformation.” The manuscript does not clearly show or support this claim.

We originally intended the discussions in the ‘Related work’. However, this section is far from the sentence. We reconsider that such a statement is not necessary in this paragraph, and rewrote to say that applications in this study can be solved with simple implementation (line 257–259).

Line 157 “However, larger-scale realistic problems with material discontinuities and rheologies require an increasing number of corresponding NNs, which incur more computational costs for...” Why more NNs? why not NNs with larger number of parameters? Please provide a more detailed discussion here.

The sentence was confusing. “More NNs” result from material discontinuities and different rheologies that are modeled by distinct NNs. We modified the sentence (line 259–261).

Reviewer #2

1) Structure. Since this is a methods paper, I recommend placing the Methods section in the main text and not in the extended information section. While reading the main text, I thought it was missing key information, such as the underlying equations and comparison to analytical solutions. This material is in the extended information, but it is central to the paper and I imagine that anyone who reads the article will want to read it as well. In terms of figure, I suggest placing Fig. 5 and perhaps Fig. 3 also in the main text.

Thank you for a valuable suggestion on the readability of the manuscript. We described outline of PINN modeling and comparison to analytical solutions to the main text, whereas left the mathematical expressions in the ‘Methods’ section. We prepared a figure explaining problem setting and NN structure (Figure 1).

2) Information on computational costs. To understand if PINNs are competitive compared to FEMs, an idea of the computational costs would be very useful: how long does training take, and how does it compare to using FEM directly? An estimate of the run times, with details of the hardware (e.g. CPUs or GPUs?) would be very valuable.

PINN modeling is generally known to take more time than common solvers for now (references 40 and 41) and we consider that development of efficient algorithms is necessary to be comparable. Instead, we compared computational time between various target problems (Table 1), which clarified which factors increase computational time significantly. We mentioned computational time in the last paragraph of the original manuscript, but discussed it in more detail in the ‘Computational costs’ section (line 234–239).

In contrast, PINN has a potential advantage in memory consumption on large-scale problems. In FEM, the number of mesh grids increases considerably with model size and dimension. In PINN, although the number of iterations should be increase considerably, NNs are trained with minibatch and training points need not be stored. Therefore, memory consumption would not increase significantly. We discussed it in line 84–90.

A related questions is the cost of training several NNs for separate spatial domains (e.g. Figure 2). Does it simply scale linearly with the number of domains? This is important to understand if one wants to model complex cases with a large number of discontinuities.

Computational cost would scale linearly with the number of domains. Experimentally, the use of two NNs increased computational time per iteration by ~ 1.6 folds (Table 1, line 226–229). Table 1 suggests that slip distributions have more influence on computational time.

3) More detail on how the model is constructed would help. For example:

* Ln. 344-351: please explain the choice of network architecture, activation functions, and batch sizes. Have you experimented with other options? Are these values standard for PINNs developed for other applications (if so, add references)? Is there any theoretical reason or a rule of thumb to choose these values?

* How do initial weights affect the time needed to train the network, and how did you choose them?

* What learning rate did you use for optimization and how did you choose it?

In this study, we fixed a basic NN design on the basis of the original work (Raissi et al., 2019). This implies stability of the performance against target problems. We consider that the search for an optimal design is important for technological progress in PINNs, but in this study, the demonstration of applicability without fine-tuning would be beneficial to state that PINNs are accessible to researchers who have a little experience on deep learning.

Initial weights: We used Xavier’s initial value in the original version. In revision, we changed to use NN parameters trained on simple models for complex models to accelerate the convergence.

Learning rate: We used standard values ($\eta=10^{-3}$, $\beta_1=0.9$, and $\beta_2=0.999$).

The details are described in the ‘Methods’ section (line 331–340).

Ln. 101-102, and figures: Please describe the length scales and other normalizations, both in the text and in figures. Values for shear modulus and strain are unphysical. Since this is linear elasticity it all works out, but it can be confusing - a brief explanation would help.

We explained the normalization in line 142–145.

Ln. 106: please describe the slip distribution for the “distributed slip” model.

We plotted the slip distributions in Figure 2b. We mentioned that the distributed slip in Model 2B is the same as Model 1C in line 166–168.

Ln. 96. “We consider a model for strike-slip faults that repeatedly cause destructive earthquakes in inland areas.”: this sounds too specific a description for your problem setup; and for a

moment I thought you were modeling seismic cycles. The fault could be creeping, or produce earthquakes that are not repeated, or not be inland. I would suggest phrasing as follows: “We consider slip on a strike slip fault, a model that could be used to describe repeated destructive earthquakes...”

We revised the sentence based on your suggestion (line 103–104).

Reviewer #3

However, I feel that one issue must be addressed for this study to be publishable. That is to consider adding an example with a quasi-static deformation problem. This is because a previous contribution (DeVries et al., 2017) used a neural network approach for this more sophisticated problem, and you could take the opportunity to express how PINNs could do better or are at least a strong alternative.

Thank you for suggesting a crucial related work. DeVries et al. (2017) applied neural networks for modeling of crustal deformation, but the target task was different from our study (Table R1). DeVries et al. (2017) trained a neural network (inputs are surface location (x, y) , time t , source depth d , and Maxwell and Kelvin viscosity η_M and η_K ; outputs are x , y , and z displacements) by generating $\sim 2 \times 10^8$ training data (layer structure and shear modulus are fixed) using a conventional semianalytical code (reference 26). The trained neural network can estimate displacements at unseen coordinates (x, y, t) and in non-simulated viscous structure (η_M, η_K) , which allows fast simulations in many parameter settings. This approach is applicable to relatively simple problems to which semianalytical methods can be applied.

In contrast, PINNs can solve complex problems without using any existing solver, although optimization is required for each parameter setting. Because the motivations of the two approaches are different, an effective comparison is difficult. In simple problems to which semianalytical methods can be applied, DeVries et al. (2017) will solve much faster than PINNs. In contrast, complex problems are the main target of PINN modeling, but these cannot be addressed by the method of DeVries et al. (2017).

DeVries et al. (2017) is a pioneering study that applied neural networks to modeling crustal deformation. We discussed their work and the difference from our study in the ‘Related work’ section (line 95–100).

A quasi-static deformation problem is an important target of PINN applications. This problem involves time dependence and many possible rheology structures, which will complicate technical implementation and augment discussions. We would like to address this problem in future work.

Table R1. Comparison between DeVries et al. (2017) and this study.

	DeVries et al. (2017)	This study
Target task	Acceleration of simulations	Solving PDEs
Conventional solver	Used	Not used
Applicable problem	Simple	Complex
Computational time	Fast	Slow

Line 35 and lines 83-85. For fully numerical methods you could also refer to Langer et al. (2019).

We added Langer et al. (2019) in the ‘Related work’ section (reference 37).

Lines 41-43. “These techniques extract latent relations from observational data without knowing the physical laws, and their applications to problems with limited data have not been successful. “ I don't quite know what you mean here, as the machine learning applications to seismic event detection have been very successful. Maybe just say, alternate applications to solving physics based problems are not as amenable to these techniques.

We reconsider the sentence, and concluded it is not essential for the discussion. We therefore removed this sentence and slightly modified the next sentence (line 38–40).

All applications appear to be 2D (infinitely long faults). How would the computational cost increase if you did 3D problems?

On the size of NN, higher dimensional problems only slightly increase input and output variables (typically less than 10 variables). Therefore, there is no problem on memory constraint, in contrast to discretization methods in which the number of meshes increases significantly. We added this discussion in the ‘Related work’ section (line 84–90). On the other hand, the computational time for PINN optimization can considerably increase because collocation points are distributed in higher dimensions. In machine learning, neural networks have been applied to much higher dimensional (> 100) problems, and PINNs have been successfully applied to 3D problems in the literature, which imply the applicability to 3D crustal deformation. Nevertheless, the convergence in a reasonable computational cost would depend on individual problems. We discussed optimization in higher dimensions (and larger model region) in the ‘Discussion’ section (line 259–263).

REVIEWERS' COMMENTS

Reviewer #2 (Remarks to the Author):

The authors made substantial changes to address the points raised by myself and reviewer #1 (which I was asked to comment upon). The new examples and figures; the discussion on computational time (and the different factors contributing to it); and the improved description of the network architecture significantly strengthen the manuscript. I only have relatively minor suggestions, in order of importance:

1) Table 1 would be more informative if it also included the computation time for existing methods (for example, the Pylith simulation used for comparison). Furthermore, it would be useful if the number of collocation points and network architecture was briefly described in the text when discussing the table, to give a sense of the problem size. (Similarly, computation time for the FEM example should be accompanied by number of elements).

2) Rather than saying that the solutions “behave similarly” (e.g. line 147, 205) please quantify the difference as a percentage; either in the text, or even better as an additional panel in Fig. 2,3.

3) The subsection named “related work” seems out of place right at the beginning of the “results” section.

Typos:

Line 145 - is a derivative with respect to z missing? ($E_{xz} = U_{x,z/2}$)

Reviewer #3 (Remarks to the Author):

Minor comment

The three fault geometries considered in Supplementary Figure 3 are not clear. The reader is referred to Figure 3a of the main text, but it shows just one fault surface.

Response to reviewers' comments on

“Physics-Informed Deep Learning Approach for Modeling Crustal Deformation”

Reviewer #2

1) Table 1 would be more informative if it also included the computation time for existing methods (for example, the Pylith simulation used for comparison). Furthermore, it would be useful if the number of collocation points and network architecture was briefly described in the text when discussing the table, to give a sense of the problem size. (Similarly, computation time for the FEM example should be accompanied by number of elements).

We measured computational time for FEM. Because FEM models were computed only for Models 2A and 2B (analytical solutions were obtained for Models 1A, 1B, 1C, 3A and 3B), we described computational time in the text instead of Table 1 (lines 212–214). The number of collocation points and network architecture in PINN modeling (lines 196–197) and the number of elements in FEM modeling (lines 214–215) were also described.

2) Rather than saying that the solutions “behave similarly” (e.g. line 147, 205) please quantify the difference as a percentage; either in the text, or even better as an additional panel in Fig. 2,3.

We calculated the root-mean-square errors between PINN and reference (analytical or FEM) solutions in Table 1 and discussed them in the text (lines 142–145; 167–168; 184–187). We calculated absolute errors because the displacements in Models 1B, 3A, and 3B cross zero.

3) The subsection named “related work” seems out of place right at the beginning of the “results” section.

The “related work” subsection was moved to the end of ‘Introduction’ and the heading was removed. Accordingly, a few sentences were modified (lines 60–66). The former descriptions of conventional methods were considerably reduced (lines 32–38) because these were discussed in the latter descriptions (lines 66–97).

Line 145 - is a derivative with respect to z missing? ($\text{Ex}_z = \text{U}_{x,z/2}$)

In fact, this expression is correct; u is the z component of displacement and u_x is its derivative with respect to x . However, we noticed that this notation (i.e. subscript represents derivative, which is used for PINN) is defined only in the ‘Methods’ section. We simply omitted the formula because it is not critical for subsequent discussions (line 140).

Reviewer #3

The three fault geometries considered in Supplementary Figure 3 are not clear. The reader is referred to Figure 3a of the main text, but it shows just one fault surface.

We revised the caption of Supplementary Figure 3 as follows:

- the fault geometries are distinguished by displacement discontinuity in the left panels;
- the surface topography and mechanical properties are the same as those in Figure 3a.